# Rhamnolipid 89 Biosurfactant Is Effective against *Streptococcus oralis* Biofilm and Preserves Osteoblast Behavior: Perspectives in Dental Implantology

**DOI:** 10.3390/ijms241814014

**Published:** 2023-09-13

**Authors:** Erica Tambone, Chiara Ceresa, Alice Marchetti, Silvia Chiera, Adriano Anesi, Giandomenico Nollo, Iole Caola, Michela Bosetti, Letizia Fracchia, Paolo Ghensi, Francesco Tessarolo

**Affiliations:** 1Department of Industrial Engineering & BIOtech, University of Trento, 38123 Trento, Italy; erica.tambone@gmail.com (E.T.); silvia.chiera@unitn.it (S.C.); giandomenico.nollo@unitn.it (G.N.); francesco.tessarolo@unitn.it (F.T.); 2Department of Pharmaceutical Sciences, Università del Piemonte Orientale “A. Avogadro”, 28100 Novara, Italy; alice.marchetti@uniupo.it (A.M.); michela.bosetti@uniupo.it (M.B.); letizia.fracchia@uniupo.it (L.F.); 3Department of Laboratory Medicine, Azienda Provinciale per i Servizi Sanitari, 38122 Trento, Italy; adriano.anesi@apss.tn.it (A.A.); iole.caola@gmail.com (I.C.); 4Department CIBIO, University of Trento, 38123 Trento, Italy; dr.ghensi@gmail.com

**Keywords:** antibiofilm activity, biocompatibility, dental implants, dislodging action, microbial biosurfactants, osteoblasts, rhamnolipids, titanium, peri-implantitis, mucositis

## Abstract

Biofilm-related peri-implant diseases represent the major complication for osteointegrated dental implants, requiring complex treatments or implant removal. Microbial biosurfactants emerged as new antibiofilm coating agents for implantable devices thanks to their high biocompatibility. This study aimed to assess the efficacy of the rhamnolipid 89 biosurfactant (R89BS) in limiting *Streptococcus oralis* biofilm formation and dislodging sessile cells from medical grade titanium, but preserving adhesion and proliferation of human osteoblasts. The inhibitory activity of a R89BS coating on *S. oralis* biofilm formation was assayed by quantifying biofilm biomass and microbial cells on titanium discs incubated up to 72 h. R89BS dispersal activity was addressed by measuring residual biomass of pre-formed biofilms after rhamnolipid treatment up to 24 h. Adhesion and proliferation of human primary osteoblasts on R89BS-coated titanium were evaluated by cell count and adenosine-triphosphate quantification, while cell differentiation was studied by measuring alkaline phosphatase activity and observing mineral deposition. Results showed that R89BS coating inhibited *S. oralis* biofilm formation by 80% at 72 h and dislodged 63–86% of pre-formed biofilms in 24 h according to concentration. No change in the adhesion of human osteoblasts was observed, whereas proliferation was reduced accompanied by an increase in cell differentiation. R89BS effectively counteracts *S. oralis* biofilm formation on titanium and preserves overall osteoblasts behavior representing a promising preventive strategy against biofilm-related peri-implant diseases.

## 1. Introduction

Dental implantology has gained high success rates enabling the effective replacement of missing teeth with satisfactory results in terms of both aesthetic and functional restoration [1,2]. However, a non-negligible fraction of implants still undergoes infective complications and possible failure due to the formation of microbial biofilm at the surface of implants and trans-mucosal implant components [3]. Biofilms are complex microbial communities adherent to abiotic or biotic surfaces, embedded in an extracellular matrix, and difficult to eradicate as highly resistant and resilient to conventional antimicrobial therapy. Biofilms on dental implants are responsible for severe associated diseases, such as mucositis and peri-implantitis, that affect the tissue around the implant, bringing to their potential failure [4,5], and influencing negatively the quality of life of patients, impairing their nutrition and wellbeing and increasing stress and anxiety [6]. Nowadays, peri-implantitis affects 22% of dental implants, with a constantly growing prevalence [7]. The most common clinical treatment is based on biofilm removal by mechanical debridement which may be associated with chemical decontamination [8]. However, the clinical outcomes of these treatments are suboptimal with no significant evidence of the usefulness of currently available chemical decontamination methods [9] indicating the need to identify new prevention and treatment approaches, able to inhibit microbial adhesion and biofilm formation on dental implants and trans-mucosal implant components as well as to remove already consolidated biofilm [10,11].

Microbial biosurfactants (BSs) are a group of natural molecules that have gained a great interest for their numerous physiochemical and biological properties. BSs have strong antimicrobial, antiadhesive and antibiofilm activity against a wide range of bacterial species, having a set of properties such as the ability to cause changes in cell membrane permeability and integrity, to promote membrane disfunction, to disrupt protein structures and to interfere with gene expression and quorum sensing signaling of microorganisms embedded within the biofilm matrix [12]. When applied as coating agents on abiotic surfaces, BSs induce changes in the physicochemical properties of the interface, such as roughness and hydrophobicity, thus making them less attractive to microbial colonization and adhesion [13,14]. BSs are also characterized by low toxicity and high biocompatibility toward eukaryotic cells, making them excellent candidates for different applications in pharmaceutical and biomedical fields [15,16,17]. Among the most promising BSs, rhamnolipids have shown the ability to exploit both strong antiadhesive characteristics and strong to moderate antimicrobial properties [18,19,20,21].

In the context of oral health and focusing on implant-associated infection and treatment, the number of the studies conducted on BSs is limited, suggesting that the application of BSs in this field is still at an early stage [22]. Rhamnolipids from *Burkholderia thailandensis* E264, mixed rhamnolipids JBR425 and lactonic sophorolipids showed a good antimicrobial activity against oral pathogens such as *Streptococcus mutans*, *Streptococcus oralis*, *Streptococcus sanguinis*, *Actinomyces naeslundii* and *Neisseria mucosa* planktonic and sessile cells. In addition, BSs demonstrated the ability to prevent the formation of oral pathogens biofilms on polystyrene both when added in free-form to a solution and when applied in the form of coating over a surface [23,24]. In 2019, Tahmourespour et al. evaluated the antibiofilm activities of a biosurfactant from *Lactobacillus rhamnosus* ATCC7469 against *S. mutans* and its effect on the expression level of the adhesion-related genes showing significant inhibitory activity on *S. mutans* adhesion and biofilm formation, with a downregulation of *gtfB/C* and *ftf* associated genes [25]. More recently, we tested for the first time the in vitro efficacy of a rhamnolipid coating on different commercial titanium surfaces against microbial colonization [21,26]. The rhamnolipid 89 biosurfactant (R89BS), composed by 70.6% of mono-rhamnolipids and 20.8% of di-rhamnolipids with a 91.4% purity grade [27], when coated to titanium, acted as an efficient antiadhesive agent, significantly inhibiting the formation of both single (*Staphylococcus aureus* and *Staphylococcus epidermidis*) and bacterial/fungal (*S. aureus*/*Candida albicans*) dual biofilms, limiting both biofilm biomass and microorganisms’ metabolic activity [21,26]. Experiments addressing the biocompatibility showed the eluate from R89BS-coated titanium discs was non-cytotoxic for both fibroblasts [21] and osteoblasts [26].

Following these encouraging results, this study further investigates the safety and efficacy of R89BS as antibiofilm agent in dental implantology. To this aim, we first evaluated the effect of R89BS on formation, growth and dispersal of *S. oralis* microbial biofilm on medial grade titanium, and, second, we characterized the impact of the R89BS coating on the adhesion, proliferation and differentiation of primary human osteoblasts on titanium.

## 2. Results

### 2.1. Antibacterial Activity on Planktonic Cells

The antibacterial activity of R89BS against *S. oralis* planktonic cells was assessed by the microdilution method. Results are summarized in Table 1, where OD_600nm_ values and percentages of growth inhibition at the different R89BS concentrations are reported. The values measured for the cells co-incubated with the rhamnolipid significantly differ from that observed for the positive control of growth, as confirmed by one-way ANOVA (*p* < 0.001). The inhibitory effect of R89BS gradually increased following a concentration-dependent trend, with a maximum at 30 µg/mL (MIC), for which no bacterial growth was observed with unaided eyes and a percentage of inhibition equal to 99% was found by the spectrophotometric analysis.

### 2.2. Effects on Bacterial Cell Surface Hydrophobicity and Membrane Permeability

Changes in *S. oralis* cell surface properties, such as surface hydrophobicity and membrane permeability, were investigated after 1 h treatment with R89BS at MIC value. The rhamnolipid altered both cell surface hydrophobicity and membrane permeability, compared to *S. oralis* untreated cells. In particular, cell surface hydrophobicity was significantly reduced from 99% (control) to 32% (treated sample) (*p* < 0.001). On the contrary, the exposure to R89BS induced a significant increase in crystal violet entrapment, from 57% (untreated control) to 79% (treated sample) (*p* < 0.001), affecting significantly bacterial cell membrane permeability.

### 2.3. Dislodging Activity on Pre-Formed Bacterial Biofilm

The ability of R89BS to dislodge *S. oralis* pre-formed biofilms on titanium surfaces was evaluated after 4 h and 24 h of co-incubation, through biomass quantification, using the CV assay.

Results are summarized in Table 2, where A_570nm_ values and percentages of removal efficacy at the different R89BS concentrations are reported. *S. oralis* biofilm biomass was significantly affected by both R89BS concentration (*p* < 0.001) and incubation time (*p* < 0.001), as confirmed by two-way ANOVA. Pre-formed biofilms were already significantly reduced by the R89BS after 4 h of co-incubation but a prolonged exposure (24 h) increased the removal of the biofilms from the discs. In particular, biofilm dislodgement rates ranged from 36% to 59% at 4 h and from 63% to 86% at 24 h.

### 2.4. Inhibitory Activity on Bacterial Biofilm Formation

The ability of TDs coated with R89BS in inhibiting *S. oralis* biofilm growth was investigated after 24 h, 48 h and 72 h of incubation by CV and viable cell counting methods.

*S. oralis* biofilm gradually grew over time, both on uncoated and R89BS-coated discs (Figure 1), but a markedly lower amount in biofilm biomass (Figure 1a) and number of CFU/disc (Figure 1b) was found in R89BS coated samples. A significant difference in terms of biomass and cell viability was present between the values measured for biofilms on coated titanium surfaces compared to those detected for uncoated controls at each experimental timepoint (Figure 1).

SEM imaging of TDs showed a markedly lower amount of bacterial cells adhering on R89BS-coated TDs with respect to uncoated TDs at all the investigated time-points (Figure 2). No morphological modification in terms of *S. oralis* cell shape and size was observed between the two experimental conditions. Three-dimensional organization of the bacterial biofilm was observed only on uncoated TDs, while bacteria were organized in a single layer at the surface of R89BS-coated titanium.

As summarized in Table 3, the coating with R89BS significantly reduced *S. oralis* biofilm formation on titanium surface up to 72 h of incubation. Biofilm biomass formed on coated discs was inhibited more than 90% (with respect to the uncoated surface) up to 48 h (*p* < 0.001) whereas it was detected a lower even if significant reduction of 61% at 72 h (*p* < 0.001). A higher impact of R89BS coating on the number of viable cells was obtained, with inhibition percentages > 98% (*p* < 0.001) up to 72 h of incubation.

### 2.5. hOBs Adhesion and Proliferation

The adhesion and proliferation behavior of hOBs on the surface of uncoated and R89BS-coated TDs was investigated using orange acridine fluorescent dye at different time-points.

Results of tests performed at 1, 3 and 24 h showed no significant differences in hOBs adhesion on R89BS-coated and uncoated TDs. Cell morphology showed similar features on both surfaces, indicating a good cell adhesion and spreading with filopodia protrusion. Representative images of hOBs adhering on R89BS-coated and uncoated TDs are shown in Figure 3a. Inspection of images collected at lower magnification revealed a homogeneous and comparable cell distribution over the titanium surface, irrespectively from the presence of the coating. At higher magnification, on both surfaces, cells showed extended filopodia, indicating a good cell-substrate interaction and no cell damage or distress.

The quantification of adherent cells obtained by cell enumeration at 1, 3 and 24 h is presented in the graph of Figure 3b. Adherent cells ranged from 72% to 95% of the initially seeded cells (1 × 10^4^ hOBs/disc). No significant differences in adhering cell count were present between R89BS-coated and uncoated TDs at any of the tested time-points, indicating no adverse effect of R89BS coating on the efficiency of the cell adhesion process.

Representative images of cell morphology and distribution at later time-points (6 and 12 days) are shown in Figure 4a, displaying comparable cell morphology and the typical flattened healthy osteoblasts shape on both tested surfaces. No sign of cell distress was visible.

A lower osteoblast cell number was observed on the R89BS-coated TDs surface at both 6 and 12 days. This evaluation (Figure 4a) was confirmed by quantitative cell counting (Figure 4b). Indeed, hOBs seeded on uncoated TDs exhibited a significantly higher proliferation rate on day 6, with a 104% increase in cell count compared to the initial number of seeded cells (2 × 10^3^ hOBs/disc). Differently, hOBs on R89BS-coated TDs showed only a 23% increase from initial cell number at the same time-point. This trend consolidated at 12 days, with uncoated TDs showing a significantly higher cell proliferation (increased by 292%) compared to R89BS-coated TDs showing an increase in cell number limited to 34%.

### 2.6. Cell Differentiation on R89BS-Coated and Uncoated TDs

The differentiative activity of human osteoblasts was assessed at 6 and 12 days on R89BS-coated and uncoated discs. At 6 days after seeding, total alkaline-phosphatase (ALP) values (Figure 5a) were significantly higher in hOBs growing on uncoated TDs with respect to R89BS-coated TDs (*p* < 0.05). This difference was no more present at 12 days when quantification of total adenosine-triphosphate (ALP) indicated no difference between the two testing conditions. Results of concomitant ATP tests (Figure 5b) confirmed the proliferation data obtained with cell enumeration (Figure 4b). The calculation of ALP expression normalized to ATP data (i.e., normalized to cells number) indicates a similar ALP single cell activity at day 6 irrespectively from the presence of the R89BS coating, and a significantly higher ALP production per single cell at day 12 for hOBs grown on R89BS-coated TDs (Figure 5c).

Test performed at 21 days from seeding, using both the standard culture medium and the same medium supplemented with dexamethasone, confirmed that no differences were present in the micro-morphological aspect of cells and their capability of differentiating toward a mature form when exposed to an appropriate stimulus (Figure 6a). Further, calcium phosphate deposits were identified by EDX analysis in hOBs at 21 days stimulated with dexamethasone, documenting their ability to create mineralized structures irrespectively from the presence of the R89BS coating (Figure 6b).

## 3. Discussion

R89BS has demonstrated its antibiofilm potential as a coating agent on different medical grade materials, including silicone, titanium and titanium alloys [21,26,28]. Antibiofilm activity was not limited to monomicrobial biofilm, but resulted effective also in case of polymicrobial biofilm, including bacterial and fungal species of clinical relevance for implant-associated infections [29]. The application of the R89BS coating on medical grade titanium has recently shown promising results for developing innovative strategies to prolong the lifespan of dental implants by limiting adhesion and reducing biofilm formation of *S. aureus* and *S. epidermidis* [21,26]. These findings opened new perspectives on mitigating the impact of infective-related peri-implant diseases such as mucositis and peri-implantitis.

In this study, the antimicrobial and antibiofilm activity of R89BS was evaluated against a different bacterial species, *S. oralis*, which has been identified as an important member of the oral microbiome, playing not only a crucial role in the maintenance of oral health but also in the development of oral diseases, such as periodontitis [30,31,32]. Indeed, *S. oralis* can significantly contribute to the formation of oral multispecies biofilms, being one of the early colonizer bacteria of the salivary pellicle-coated surfaces in the oral cavity [33,34].

The antibacterial activity of R89BS in aqueous solution resulted to be concentration dependent. A concentration of 30 μg/mL inhibited 99% of *S. oralis* growth indicating a strong antimicrobial effect of this compound at concentrations that are non-cytotoxic for eucaryotic cells [21]. The chemic-physical action of the R89BS on *S. oralis* cell wall was also documented, by the significant reduction in cell surface hydrophobicity and the increase in cell membrane permeability. In principle, these characteristics allow R89BS to effectively interfere with cell–cell and cell–surface contact mechanisms, reducing the hydrophobic interactions that cause the bacteria to initially adhere to the implant surface [35]. It is known that rhamnolipids BS can interact with the phospholipidic bilayer of bacterial membrane altering membrane integrity and protein structure, interfering in the electron transport chain and energy generation and can also increase membrane permeability through the release of lipopolysaccharide and the formation of transmembrane pores, causing a loss of intracellular metabolites and cell lysis [12,15]. Moreover, it was observed that the treatment with rhamnolipids modifies cell surface composition, solubilizing and releasing apolar components causing a reduction in surface hydrophobicity [36].

The effect of R89BS was also tested on preformed biofilms on titanium discs for a short period of 4 h and for a prolonged exposure of 24 h to further increase the removal of biofilms from the discs. Previous studies reported about the good disrupting efficacy of rhamnolipid BS on pre-formed biofilm, recognizing the alteration of the extracellular matrix and an impairment of the biofilm structure stability [28,37,38,39]. In this study, the biofilm dislodgement rates of R89BS ranged from 36% to 59% at 4 h and from 63% to 86% at 24 h, demonstrating the effectiveness of R89BS in dislodging pre-formed biofilms of *S. oralis* from titanium surfaces. The results also emphasized the impact of incubation time on the efficacy of R89BS. The longer exposure of 24 h led to a higher rate of biofilm removal, indicating that extended treatment duration can enhance the biofilm-dislodging properties of R89BS. Although these in vitro data require additional investigation using polymicrobial biofilm, R89BS deserves potential as a decontaminating agent to assist debridement procedures in the conservative treatment of peri-implantitis, currently lacking effective solutions [8,9,40].

Remarkable results were finally obtained in tests addressing the inhibitory effect of R89BS coating on de novo formation of *S. oralis* biofilm. Significant differences were observed in terms of biomass and cell viability between the *S. oralis* biofilms formed on coated titanium surfaces and those on uncoated controls up to 3 days. The coating with R89BS resulted in more than 90% inhibition of biofilm formation on titanium surfaces up to 48 h, with a lower but significant reduction of 61% at 72 h. In the same experimental conditions, R89BS had a higher impact on the number of viable cells, with inhibition percentages exceeding 98% (*p* < 0.001) up to 72 h of incubation. Differences between inhibition percentages obtained between biofilm biomass and cell viability could be related to the fact that the crystal-violet staining, used for biomass determination, quantifies live cells, dead cells and extracellular matrix of the biofilm, without distinction. Further, surviving microbial cells exposed to the biosurfactant possibly increase the extracellular matrix production, as previously observed in response to treatment with antibiotics, and as a general defense mechanism to chemical insults, being the matrix the main responsible for the biofilm’s long-term survival under adverse conditions [41]. Overall, inhibition outcomes are in agreement with the hypothesis that R89BS effectively alters the surface properties, inhibiting effectively the attachment and growth of bacterial biofilms.

Intriguingly, these changes in attachment and growth of bacterial biofilm did not reflect into a detrimental interaction with eukaryotic cells. Different BSs have demonstrated to possess a positive effect on bone tissue formation. Surfactin, a lipopeptide biosurfactant, proved to induce higher expression of ALP and to increase the calcium deposit in an in vitro culture of bone marrow mesenchymal stem cells [42]. Moreover, it reduced the osteogenic differentiation of bone marrow macrophages, exerting a preventive activity against bone loss [42]. BSs have been also added to bioglasses and cements used in bone regeneration suggesting that they could be a valuable factor with different activities on microorganisms, inflammatory microenvironment and eukaryotic cells [43,44].

In a previous study, we investigated the cytotoxic effect of R89 biosurfactant on hOBs proving that concentrations up to 50 µg/mL in the culture medium did not compromise cell viability [26]. Furthermore, no cytotoxic effect was observed on hOBs cultured in the eluate obtained from titanium discs coated with R89BS [26]. In the present work, we performed in vitro tests to evaluate the effect of R89BS on hOBs growing in direct contact with a R89BS-coated titanium surface. hOBs behavior was characterized by comparing adhesion/proliferation of cells and their differentiation activity on coated and uncoated titanium discs. Orange acridine fluorescent images taken at an early time-point of 3 h did not show any sign of cell distress presenting a similar cell morphology and spreading on the discs surface irrespectively from the presence of the R89BS coating. The good cell adhesion we qualitatively observed using optical microscopy, was confirmed by direct cell count, presenting comparable numbers of adherent cells at 1 h, 3 h and 24 h after seeding. Cellular adhesion is a key mechanism that paves the base for osseointegration and regulates many aspects of hOBs behavior such as cell survival, proliferation and differentiation. Adhesion starts with the physical attachment of cells to the surface, followed by spreading [35,45]. A range of molecules are involved in the adhesion process at the interface between the cell membrane and the surface. The presence of the R89BS coating is supposed to have a major impact on these molecular processes, but unfortunately, at present, no studies have investigated in detail the involvement of rhamnolipids in the adhesion of eukaryotic cells. Possibly also micromorphological aspects (e.g., changes in the surface roughness induced by the coating) have to be considered in osteoblasts adhesion given the recognized importance of the relation between surface roughness and adhesion of human primary bone cells [45]. BSs are widely used because of their non-attachment properties for prokaryotic cells [46], and our results indicate that no relevant impairment in the adhesion process was present in our experimental conditions for eukaryotic cells.

Longer time-points were investigated to understand late effects of R89BS coating on hOBs in terms of proliferation and differentiation. Qualitative evaluation of cell morphology confirmed that no sign of cells distress was present at any of the tested time-points, in agreement with [47]. However, both direct cell enumeration and ATP quantification at day 6 and at day 12 evidenced a reduction in hOBs proliferation on R89BS-coated titanium with respect to the uncoated samples. Together with the reduction in hOBs proliferation, a decrease in the total ALP expression was observed on R89BS coated titanium at 6 days but not at 12 days. ALP data, normalized to the number of hOBs on R89BS-coated TDs, indicated no difference in single cell ATP production at day 6 and a higher ALP single cell production at day 12. These preliminary results, according to [48,49], suggest that the rhamnolipid coating does not reduce cell viability, but rather favors the transition to mature osteoblasts while moderately reducing cells proliferation [49]. The evolution of cell morphology from elongated-fused fibroblast-like cells to mature trapezoidal osteoblasts was observed at days 6, 12 and 21. The morphological changes in adult primary osteoblasts have been previously related to cell proliferation or differentiation [47]. Similarly, anisotropic surfaces, inducing low surface enlargement, generally favored cell proliferation, whereas the proliferation rate decreased on surfaces with increased isotropy [50]. Moreover, in osteoblasts, when proliferation ceases, alkaline phosphatase levels increase and then an ordered deposition of minerals is initiated within the extracellular matrix of these nodules resulting in the development of a bone tissue-like organization [47]. In this study, a qualitative assessment of cell morphology was obtained by fluorescence imaging taken at 6 and 12 days after seeding. For both experimental conditions, hOBs grown on R89BS-coated and uncoated TDs showed a flattened morphology comparable to healthy osteoblasts. At 12 days, R89BS decreased hOBs proliferation not impairing cell viability, an effect that can be explained by osteoblast differentiation induction. Indeed, hOBs in proliferative phase have high ATP rate and low differentiative markers such as ALP, osteopontin, collagen type I or bone sialoprotein [48], which conversely are highly expressed when the osteoblast is in the differentiative phase alongside with a decrease in ATP production. The formation of cell-induced mineral deposits on titanium was investigated by SEM-EDXS. On both coated and uncoated samples, mineral formation was microscopically observed at 21 days of culture. Composition of mineral deposits was investigated by elemental analysis demonstrating the presence of calcium and phosphorus. Mineral deposits were not observed in the absence of cells, strongly suggesting that the mineral deposits were produced by hOBs. Overall, morphological and compositional data indicate no compromission of the hOBs’ differentiative process in presence of R89BS, when proper stimuli are provided (e.g., dexamethasone).

### Study Limitations

Despite obtained results are encouraging, several aspects remain to be evaluated. Among these, the efficacy of BSs against complex polymicrobial biofilm, such as those typical of the oral cavity, has still to be addressed. Immediately after a pellicle of proteins from saliva has adsorbed on implant surface [51,52], primary colonizer microorganisms, such as *S. oralis* [51], express surface adhesins and bind to implant components and peri implant tissues. Biofilm formation progress by co-aggregation of a number of different additional species, depending on the surface properties, nutrient quality and availability, oxygen levels and microbial interactions including intra and inter-regnum interactions [53]. Although polymicrobial in-vitro models are reported in the literature [53,54,55], major technical limitations occur in mimicking the complexity of the peri-implant microbiota using reliable and reproducible in vitro biofilm models, encompassing the stringent culturing conditions of most of the oral microorganisms [56,57] and moving from two-dimensional to novel three-dimensional biofilm models [58]. Overall, further studies are needed to evaluate the long-term effectiveness and safety of R89BS coating in more complex environments, in in vivo conditions and finally in clinical settings. Additional confirmatory experiments should also complement and confirm the preliminary data on hOBs differentiation reported here.

The methodological approach implemented in this study should be extended to different titanium surface morphologies, including rough finishing such as those obtained by sand blasting, acid etching, laser ablation and their combinations, which are frequently applied in commercially available implants [59,60]. Results of previous studies showed that equivalent biofilm inhibition was obtained using R89BS coating, irrespectively on surface morphology, on three different commercial surface finishing, including both rough and smooth morphologies [21]. Although the preservation of the rough, bone-contacting implant surface is mandatory, the relevance of smooth titanium finishing in limiting colonization of the implant should not be underestimated. Smooth finishing are widely used in coronal implant portions and transmucosal components, playing a key-role in favoring soft tissue integration, generation of effective mucosal seal and limiting microbial penetration toward apical areas of the implant [61].

From a clinical perspective, the antibiofilm efficacy up to 72 h against *S. oralis* documented here, summed to previous results indicating similar efficacy on biofilm of other bacterial and fungal species [21,26], let foresee a relevant impact of the R89BS coating in safeguarding the implant in the short term after surgical placement. However, an extended efficacy to prevent the development of peri-implant diseases during the later stages is highly desirable and could require further development in terms of coating stability and durability at the titanium surface [21].

## 4. Materials and Methods

### 4.1. Streptococcus oralis Growth Conditions

Biofilm-producer *S. oralis* DSMZ 20379 was stored at −80 °C in Brain Heart Infusion Broth (Scharlab Italia S.r.l., Milan, Italy) enriched with 1% glucose (Scharlab Italia S.r.l., Milan, Italy) (BHI + 1%Glucose) and supplemented with 25% glycerol. The strain was grown on Brain Heart Infusion Agar (Scharlab Italia S.r.l., Milan, Italy) for 72 h at 37 °C and further processed according to Elshikh et al. [24]. In detail, a few colonies were inoculated into 40 mL of BHI + 1%Glucose and incubated at 37 °C for 16–20 h. The culture was centrifuged for 15 min at 4000 rpm, the pellet was washed twice with phosphate buffer saline (PBS) and finally resuspended in the adequate medium, according to the experimental protocols detailed below.

### 4.2. Biosurfactant Production

Rhamnolipids-producing *Pseudomonas aeruginosa* 89 was grown on Tryptic Soy Agar (Scharlab Italia S.r.l., Milan, Italy) for 16–20 h at 37 °C. The production, extraction and chemical characterization of the R89BS was performed as reported in Ceresa et al. [28]. Briefly, some colonies from the overnight culture were grown in 40 mL of Nutrient Broth II (Sifin Diagnostics GmbH, Berlin, Germany) at 37 °C for 4 h at 140 rpm. An amount of 24 mL milliliters of the resulting bacterial suspension was added to 1.2 L of Siegmund–Wagner medium and incubated at 37 °C for five days at 120 rpm. The culture was then centrifuged (Sorvall RC-5B Plus Superspeed Centrifuge, Fisher Scientific Italia, Milan, Italy) at 7000 rpm for 20 min and the supernatant collected. After acidification at pH 2.2 with 6M H_2_SO_4_ and storage overnight at 4 °C, R89BS was finally extracted three times with ethyl acetate (Merck KGaA, Darmstadt, Germany) and the organic phase was evaporated to dryness under vacuum conditions. The composition of R89BS by mass spectrometry analysis was previously reported in Allegrone et al. [27].

### 4.3. Antibacterial Activity of R89BS on Planktonic Cells

The susceptibility of *S. oralis* planktonic cells to R89BS (final concentrations ranging from 1.87 to 30 µg/mL) was determined according to the broth microdilution method described by Wiegand et al. [62], with some modifications. Briefly, 100 µL of double-concentrated R89BS solutions, freshly prepared in PBS, were added into 96-well plates and mixed with 100 µL of a *S. oralis* suspension at the concentration of 10^6^ Colony Forming Unit per mL (CFU/mL), prepared in double-strength (2×) BHI+1%Glucose. As a positive control of growth, wells were filled with the same volume of bacterial inoculum and PBS. Blank wells, containing an equal volume of BHI + 1%Glucose 2× and PBS, were also included. Plates were incubated at 37 °C for 16–20 h.

The minimal inhibitory concentration (MIC) of R89BS was defined as the lowest concentration of the rhamnolipid that prevents the visible growth of *S. oralis*. Moreover, the optical density (OD) at 600 nm was measured for each well (Ultramark Microplate Imaging System, Bio-Rad Laboratories S.r.l., Segrate, Italy), data were normalized to the blank value, and the percentages of bacterial growth reduction were calculated according to [24], using the following formula:$$Growth\;reduction\;\left(\%\right)=\left(1-\frac{OD_{R89BS}}{OD_{control}}\right)\times 100$$
where *OD_R89BS_* is the optical density mean value of samples treated with the different concentrations of R89BS and *OD_control_* is the optical density mean value of the positive control of growth. Assays were performed in triplicate and repeated in three different experimental sessions.

### 4.4. Effects of R89BS on Bacterial Cell Surface Hydrophobicity

*S. oralis* cells surface hydrophobicity after R89BS treatment was evaluated by measuring bacterial adhesion to hexadecane according to Rosenberg et al. [63]. Suspensions with an OD_600nm_ = 0.5 were prepared in PBS and co-incubated with R89BS (at MIC) at 37 °C for 1 h at 150 rpm. Bacterial suspensions without treatment were used as negative control. After centrifugation for 15 min at 4000 rpm bacterial cells were resuspended in phosphate urea magnesium (PUM) buffer and the OD_550nm_ of the suspensions was measured (Genova Plus, Jenway, UK). An amount of 4 mL of bacterial suspensions was then mixed with 1 mL of hexadecane (Scharlab Italia S.r.l., Milan, Italy) in a glass tube, stirred at high speed for 2 min and left to rest for 10 min. Afterward, OD_550nm_ of the aqueous phases was measured and surface hydrophobicity was calculated as described in [29]. Assays were carried out in triplicate and repeated in two different experimental sessions.

### 4.5. Effects of R89BS on Bacterial Cell Membrane Permeability

The permeability of *S. oralis* cell membrane after R89BS treatment was determined by checking the increased penetration of the hydrophobic dye crystal violet (CV) in cells, as proposed by Sana et al. [64]. Suspensions with an OD_600nm_ equal to 0.5 were prepared in PBS and co-incubated with R89BS (at MIC) at 37 °C for 1 h at 150 rpm. Bacterial suspensions without treatment were used as negative control. After centrifugation for 15 min at 4000 rpm bacterial cells were resuspended in a crystal violet solution (10 µg/mL) (Merck KGaA, Darmstadt, Germany) and incubated at 37 °C for 30 min at 150 rpm. Afterward, the suspensions were centrifuged at 4000 rpm for 15 min and the absorbance at 590 nm of the supernatants was measured. The percentage of entrapped crystal violet was estimated as indicated in [29]. Assays were carried out in triplicate and repeated in two different experimental sessions.

### 4.6. Medical-Grade Titanium Discs Preparation

Titanium alloy Ti6Al4V (medical-grade 5) discs (TDs) 10 mm in diameter and 2 mm in thickness were obtained from computer numerical control machining and subsequently polished with increasing fine-grained silicon-carbide abrasive paper up to 4000 grit to obtain a mirror surface. Before use for testing with bacterial and eukaryotic cells, TDs were cleaned and disinfected as reported in Tambone et al. [21]. Briefly, TDs were sonicated for 15 min each in three consecutive solutions (acetone, 70% *v*/*v* ethanol in deionized water and deionized water), to remove impurities and grinding residues, and then disinfected by immersion for at least 24 h in 70% *v*/*v* ethanol in water. To prevent microbial contamination, TDs were stored in 70% *v*/*v* ethanol in deionized water and dried in 24-well plates under a laminar flow immediately before testing.

### 4.7. R89BS Dislodging Activity on Pre-Formed Bacterial Biofilm

*S. oralis* suspensions were prepared in BHI + 1%Glucose and cell density was adjusted to 10^7^ CFU/mL. TDs were placed in sterile 24-well polystyrene plates, completely submerged with 1 mL of bacterial suspension, and incubated for 24 h at 37° C. Biofilms formed on TDs surfaces were then exposed to different concentrations of R89BS (final concentrations ranging from 30 µg/mL to 120 µg/mL in BHI + 1%Glucose), and incubated for 4 h and 24 h at 37 °C. A set of not-treated biofilms were included as positive control of growth. In addition, as blank, a number of not-treated and R89BS-treated TDs was dipped in sterile medium and incubated at the same culturing conditions. At the end of the incubation periods, the supernatants were removed and biofilms were gently washed twice with PBS.

The dislodging activity of R89BS against *S. oralis* pre-formed biofilm was evaluated by biomass quantification using the crystal violet (CV) staining method as described in [21]. Briefly, biofilms/blank TDs were air-dried and then stained with 1 mL of CV solution (0.2% *w/v*) for 10 min. Biofilms/blank TDs were washed with deionized water to remove dye excess and then air-dried overnight. Finally, the CV bound to the biofilms/blank TDs was solubilized with 1 mL of acetic acid (33% *v*/*v*; Scharlab Italia S.r.l., Milan, Italy) and spectrophotometrically determined by measuring the absorbance at 570 nm (VICTOR^3^V™, PerkinElmer Inc., Waltham, MA, USA). Data were normalized to the blank values and the percentage of biofilm biomass inhibition was estimated as indicated in [21]. Assays were conducted in triplicate and repeated in two different experimental sessions.

### 4.8. Titanium Surface Coating Process

The surfaces of TDs were coated with R89BS by physical adsorption, using the procedure described in [21]. Briefly, TDs, placed in 24-well polystyrene plates, were dipped in 1 mL of R89BS (4 mg/mL in PBS) and incubated at 37 °C for 24 h at 70 rpm. Afterward, the discs were transferred into new 24-well polystyrene plates and dried under a laminar flow to set the BS coating at the surface.

### 4.9. Antibiofilm Activity of R89BS-Coated TDs

*S. oralis* suspensions were prepared in BHI + 1%Glucose and cell density was adjusted to 10^7^ CFU/mL. R89BS-coated and uncoated TDs (as positive control of growth) were placed in sterile 24-well polystyrene plates, submerged with 1 mL of bacterial suspension, and incubated up to 72 h at 37 °C. In addition, as a blank, a set of R89BS-coated and uncoated TDs was dipped in sterile medium and incubated at the same culturing conditions. Every 24 h, fresh medium was provided to sessile cells by transferring TDs into new plates containing 1 mL of BHI + 1%Glucose. At the end of the incubation time, the supernatants were removed, and biofilms were gently washed twice with PBS.

Biofilms were quantified by CV staining and viable cell counting method as described in [26]. Briefly, TDs were placed in 50 mL tubes containing 10 mL of NaCl solution (0.9% *w*/*v*) (Sigma-Aldrich, St. Louis, MO, USA) and sessile cells were detached from the titanium surface with three cycles of sonication and stirring, for 30 s each. The resulting suspensions were serially diluted 1:10 *v*/*v* in NaCl solution (0.9% *w*/*v*). An aliquot of 100 µL of each dilution was plated on BHI agar plates. Plates were incubated at 37 °C for 24 h and colonies were then manually enumerated. Data were expressed as log_10_ CFU/disc and the inhibition percentage of biofilm formation was determined as indicated in [26]. Assays were performed in quadruplicate and repeated in three different experimental sessions.

A qualitative micro-morphological analysis of *S. oralis* biofilms formed on TDs at the different time-points was performed by scanning electron microscopy (SEM), according to the protocol described in [26]. Briefly, biofilms were immersed in 1 mL of glutaraldehyde solution (2.5% *w*/*v*; Scharlab Italia S.r.l., Milan, Italy) in 0.1 M N-(2-Hydroxyethyl)piperazine-N′-(2-ethanesulfonic acid) (Sigma-Aldrich, Milan, Italy) buffer at 4 °C. After 24 h, each biofilm was washed twice with Milli-Q^®^ water, dehydrated by immersion in 70%, 90% and 100% *v*/*v* ethanol/water solutions for 10 min each, and finally dried overnight under a laminar flow cabinet. Afterward, dried samples were coated with Pt/Pd alloy (80/20) using a Q150R sputter coater (LOT-Quantum Design, Germany) to improve their electrical conductivity and thermal stability. SEM observations were performed using a FE-SEM in high-vacuum mode (Zeiss supra 40, Carl Zeiss, Oberkochen, Germany). The primary beam energy was regulated from 3 to 5 keV to minimize the damage to the biological structures. Possible artifacts due to sample preparation were taken into consideration according to previous indications [65,66] and experiences in imaging microbial biofilm on dental implants and transmucosal components [67,68].

### 4.10. Isolation of hOBs from Human Trabecular Bone Fragments

Human primary osteoblasts (hOBs) were obtained from human trabecular bone fragments provided by the Orthopaedic and Traumatology Unit of the hospital “Maggiore della Carità”, Novara, Italy. In accordance with Rec(2006)4 guidelines of the Committee of Ministers of the Council of Europe to member states on research on biological materials of human origin [69], each patient signed an informed consent form before the collection of the biological materials, agreeing that the leftover tissue intended for disposal was made available for research purposes.

Collected trabecular bone fragments were washed in PBS and then digested in collagenase/elastase as previously described in [70]. hOBs grew from the digested bone fragments within ten days and formed a confluent monolayer on a culture dish in 3–4 weeks. hOB cells were characterized by checking osteoblastic morphology, alkaline phosphatase (ALP) expression and hormone responsiveness (PTH, 1.25(OH)_2_-D3) as previously reported in [71]. Cells were cultured in Iscove’s modified Dulbecco’s medium (IMDM, Euroclone, Milan, Italy) supplemented with 10% foetal bovine serum (FBS, Hyclone GE Healthcare, Chicago, IL, USA), penicillin/streptomycin (50 U/mL and 15 μg/mL) and 2 mM L-glutamine (L-Glu) at 37 °C in 5% CO_2_ and used within the eighth passage. Where not specified, reagents were from Sigma-Aldrich (Milan, Italy).

### 4.11. hOB Adhesion and Proliferation Tests

hOBs were cultured on uncoated and R89BS-coated TDs to evaluate and compare cell adhesion, morphology and proliferation at different time-points. Experimental protocol was adapted from [72]. More in details, R89BS-coated and uncoated TDs were placed in a 48-well microplate and 1 × 10^4^ hOBs/disc (in 1 mL of culture medium per each well) were seeded for experiments with a short time-points (1, 3 and 24 h). Differently, 2 × 10^3^ hOBs/disc (in 1 mL of culture medium per each well) were seeded for longer time-points experiments (6 and 12 days). When samples achieved the intended time-points, the culture medium was discarded, the TDs were washed with PBS, adhering hOBs were fixed for 20 min at 60 °C, and finally stained with 1 mL of orange acridine solution (2.5% in PBS) for 3 min at room temperature in dark conditions. Stained samples were washed with PBS and distilled water, and dried at room temperature before imaging adherent hOBs under a fluorescence optical inverted microscope (DMRB, Leica Microsystems, Milan, Italy). A set of representative images, at a magnification ranging from 10× to 40×, were collected to qualitatively assess hOBs morphology and distribution on R89BS-coated and uncoated TDs at each time-point. To assess cell proliferation, cell number was evaluated on eight random fields of view (having an area of 0.162 mm^2^ each) collecting the images using a 20× objective. Experiments were carried out in triplicate. Data were expressed as cells/disc considering the area of the disc equal to 78.5 mm^2^.

### 4.12. hOBs Differentiation Tests

To assess the differentiation fate of hOBs growing on R89BS-coated and uncoated TDs, a total of 5 × 10^3^ hOBs/disc were seeded and cultured for 6 and 12 days to quantify ALP activity and ATP levels. When samples achieved the intended time-points, culture medium was removed, hOBs on TDs were washed three times with Tris HCl 0.05 M, pH 7.4, and a drop of 150 µL of lysis pre-heated solution (SDS 0.05% in Tris HCl 0.05 M pH 8) was placed on top of each disc and recovered in 10 min. Cell lysate was used to quantify both ALP and ATP. In details, ALP activity, a marker of osteoblast differentiation, was quantified in 100 µL of cell lysate added to 100 µL of substrate solution (5 mg/mL of p-nitrophenyl phosphate substrate, MgCl_2_ 10 mM in Tris HCl 0.25 M, pH 9.5). After 3 h of incubation in dark and humidified conditions at 37 °C the absorbance of p-nitrophenol formed was read at 450 nm using a spectrophotometer (VICTOR^3^V™, PerkinElmer Inc., Waltham, MA, USA). ATP level, correlating with cell number, was assessed using “ATP ViaLight™” kit (Lonza Bioscience, Basel, Switzerland), adding 25 µL of cell lysate to 50 µL of Tris HCl 0.05 M and 75 µL of luciferase which catalyzes the formation of light from ATP and luciferin. Light produced, that was linearly related to the intracellular ATP concentration, was measured by a luminometer and expressed as relative luminescence units (RLUs). Experiments were carried out twice in duplicate.

The micro-morphology of the adherent osteoblasts was qualitatively assessed by scanning electron microscopy (SEM). An amount of 2 × 10^3^ hOBs/disc was seeded on uncoated and R89BS-coated TDs and cultured for 21 days and grown in culture medium with and without dexamethasone (10 mM) used as positive control. When the time-point was achieved, culture medium was removed, samples were washed with cacodylate buffer (pH 7.4) and fixed in Karnowsky’s solution (4% paraformaldehyde + 2.5% glutaraldehyde in 0.1 M cacodylate buffer pH 7.4) for 30 min at 4 °C. After washing in cacodylate buffer, cells were dehydrated in ethanol (50–100%) and then in hexamethyldisilazane [73].

Samples were observed without a metallic layer coating, to avoid interference in elemental composition determination. The observations were performed using a Quanta 200 (FEI-Philips, Eindhoven, The Netherlands) SEM in the high-vacuum mode at 10 keV beam energy. Relevant chemical composition was evaluated by energy dispersive X-ray spectroscopy (EDXS, Thermo Fisher Scientific Inc., Waltham, MA, USA) using an electron beam energy of 15 keV to address qualitatively the presence of the elements calcium and phosphorus, characterizing the mineral matrix deposited by mature osteoblasts.

### 4.13. Data Analysis and Statistics

The single TD was considered as the statistical unit. Quantitative data were expressed as mean value and standard deviation of replicated measurements after checking for normality of data distribution using Shapiro–Wilk test.

The effects of the different concentrations of R89BS on *S. oralis* planktonic and sessile cells, in comparison to uncoated controls, were evaluated with one-way ANOVA and two-way ANOVA followed by Sidak post-hoc test, respectively. A two-sample *t*-test was used to evaluate the significance of data in cell surface hydrophobicity, membrane permeability and antibiofilm assays. Results of adhesion, proliferation and differentiation tests were analyzed using two-way ANOVA followed by Sidak post-hoc test. All analyses used two-sided tests with a significance level of *p* < 0.05. Statistical tests were performed using Prism 9 software (GraphPad Software, San Diego, CA, USA).

## 5. Conclusions

R89BS effectively counteracts *S. oralis* biofilm formation on medical grade titanium and preserves overall osteoblasts behavior representing a promising preventative strategy against biofilm-related oral diseases. This microbial biosurfactant is also able to disruption 24 h old *S. oralis* biofilm from a titanium surface deserving potential as decontaminating agent in the treatment of implants affected by mucositis and peri-implantitis.

## Figures and Tables

**Figure 1 ijms-24-14014-f001:**
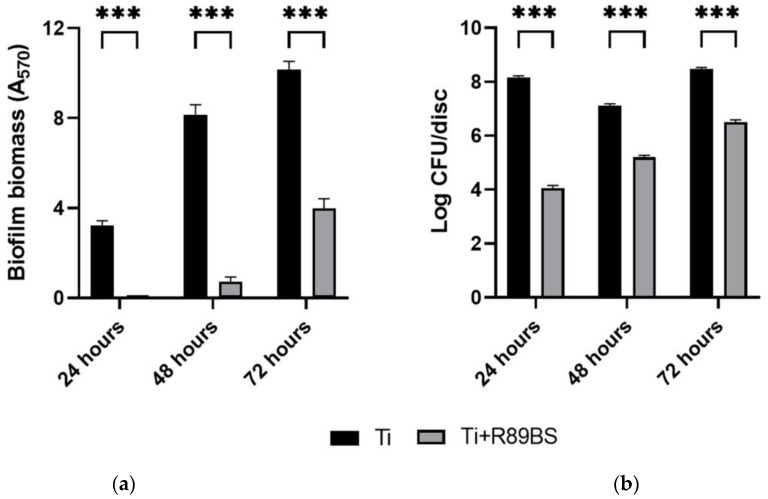
*S. oralis* DSMZ 20379 biofilm formation on R89BS-coated TDs (Ti+R89BS) with respect to uncoated controls (Ti). Results are presented in terms of biomass (**a**) and cell viability (**b**), obtained with CV staining and viable cell counting methods respectively, after 24 h, 48 h and 72 h. Error bars represent standard deviation. *** *p* < 0.001.

**Figure 2 ijms-24-14014-f002:**
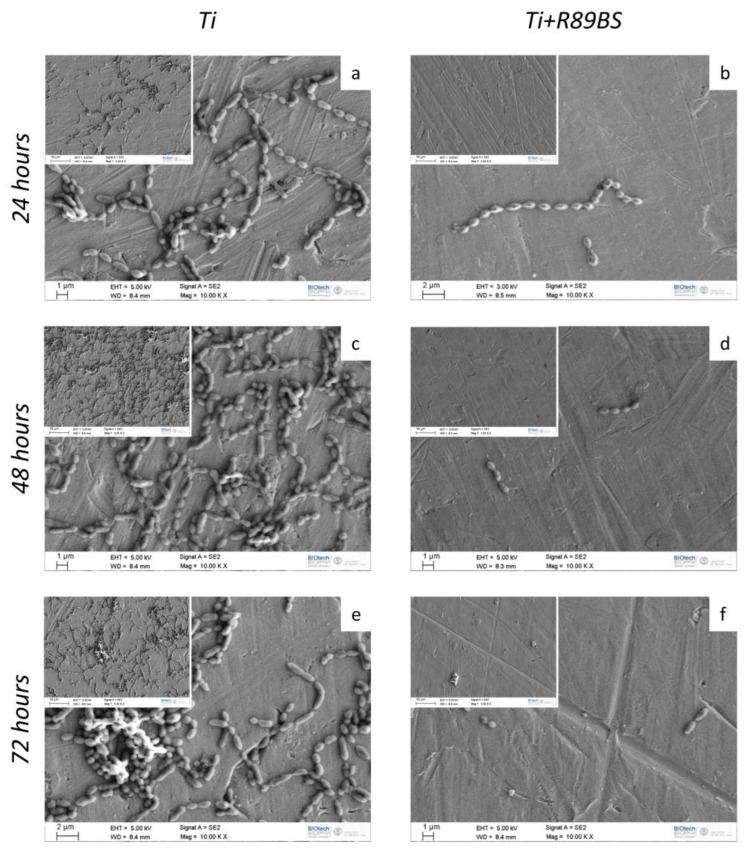
*S. oralis* biofilm grown on uncoated control TDs (Ti, panels (**a**,**c**,**e**)) and R89BS-coated TDs (Ti+R89BS, panels (**b**,**d**,**f**), after 24 h, 48 h and 72 h of incubation. Representative images were obtained by scanning electron microscopy in high-vacuum mode. Original magnification: 10,000× (insets 4000×).

**Figure 3 ijms-24-14014-f003:**
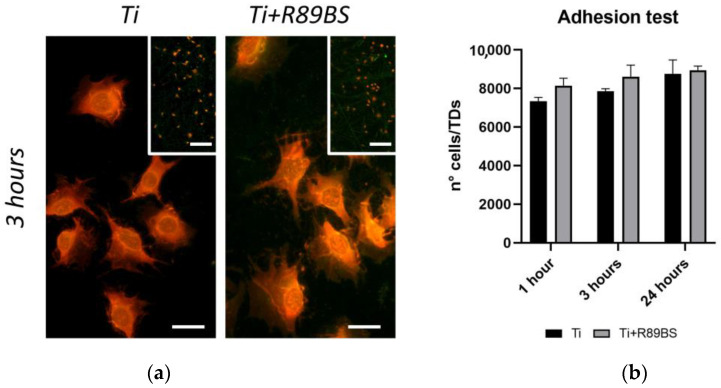
Results of adhesion tests with human primary osteoblasts seeded on R89BS-coated (Ti+R89BS) and uncoated (Ti) discs. (**a**) Representative fields of view acquired in fluorescence optical microscopy. Images were taken at 3 h post-seeding at 1 × 10^4^ cells/disc. Acridine orange stain. Original magnification 40×, bar is 20 µm (10× inset, bar is 200 µm). (**b**) Number of cells per disc at 1 h, 3 h and 24 h after seeding obtained by direct cell enumeration. Error bars represent standard deviation.

**Figure 4 ijms-24-14014-f004:**
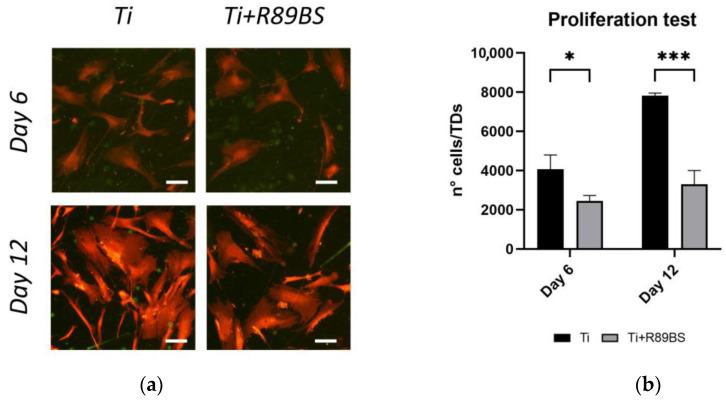
Results of proliferation tests with human primary osteoblasts seeded on R89BS-coated (Ti+R89BS) and uncoated (Ti) discs. (**a**) Representative fields of view acquired in fluorescence optical microscopy. Images were taken at 6 and 12 days after seeding at 2 × 10^3^ cells/disc. Acridine orange stain. Original magnification 10×, bar is 100 µm. (**b**) number of cells per discs at 6 and 12 days after seeding obtained by direct cell enumeration. Error bars represent standard deviation. * *p* < 0.05; *** *p* < 0.001.

**Figure 5 ijms-24-14014-f005:**
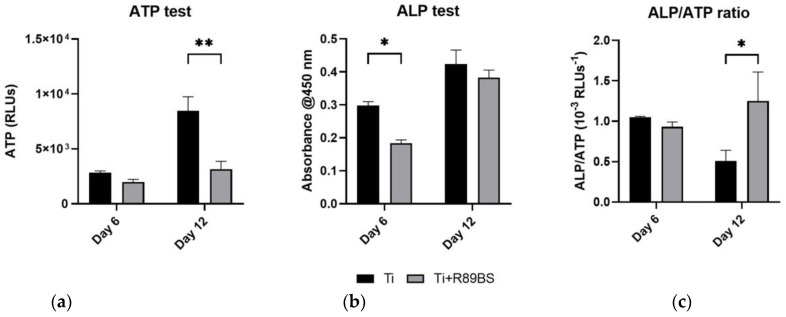
hOBs differentiation assessment. 5 × 10^3^ cells/disc were seeded on R89BS-coated (Ti+R89BS) and uncoated discs (Ti). At day 6 and 12 ALP (**a**) and ATP (**b**) quantification were performed. ALP/ATP ratio was calculated (**c**). Error bars represent standard deviation, * *p* < 0.05. ** *p* < 0.01.

**Figure 6 ijms-24-14014-f006:**
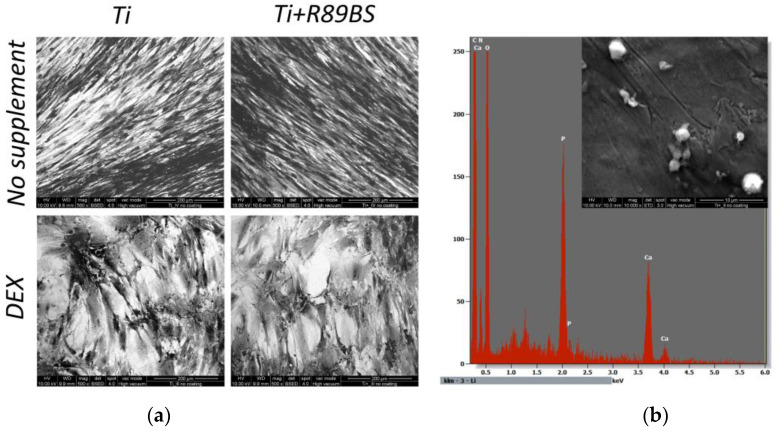
Results of differentiation tests with human primary osteoblasts after 21 days from seeding. (**a**) Morphological appearance of hOBs on uncoated and R89BS-coated TDs, and cultured in medium with (DEX) and without (no supplement) dexamethasone. Representative fields of view acquired at scanning electron microscopy using backscattered electron signal on samples without metallic coating. Original magnification 500×. (**b**) Elemental composition of calcium-phosphate deposits present on both uncoated and R89BS-coated TDs, when hOBs were cultured using dexamethasone-added culture medium. EDX spectrum was collected using a 15 keV electron beam. Image was collected using secondary electron signal. Original magnification 10,000×.

**Table 1 ijms-24-14014-t001:** Effect of different concentrations of R89BS on *S. oralis* DSMZ 20379 planktonic cells. Results are expressed as OD_600nm_ mean values and standard deviations (SD). The percentages of growth inhibition, with respect to untreated controls (0 µg/mL), are also reported.

R89BS (µg/mL)	Planktonic Cells Concentration OD at 600 nm (Mean ± SD)	Growth Inhibition (%)
0.0	0.668 ± 0.031	-
3.8	0.568 ± 0.030	15
7.5	0.442 ± 0.029	34
15	0.176 ± 0.028	74
30	0.007 ± 0.001	99

**Table 2 ijms-24-14014-t002:** Dislodging activity of R89BS on *S. oralis* DSMZ 20379 pre-formed biofilms, after 4 h and 24 h. Results are expressed as A_570nm_ mean values and standard deviations (SD). The percentages of biofilm removal, with respect to untreated controls (0 µg/mL), are also reported.

R89BS (µg/mL)	4 h		24 h	
	Biofilm Biomass (Mean ± SD)	Removal Efficacy (%)	Biofilm Biomass (Mean ± SD)	Removal Efficacy (%)
0.0	5.77 ± 0.38	-	8.82 ± 0.73	-
30	3.67 ± 0.40	36	3.26 ± 0.63	63
60	3.60 ± 0.67	38	2.86 ± 0.44	68
120	2.38 ± 0.39	59	1.20 ± 0.30	86

**Table 3 ijms-24-14014-t003:** Inhibition percentages of *S. oralis* biofilm formation determined by CV staining (biomass) and viable cell counting (cell viability), after 24 h, 48 h and 72 h. Data are reported as mean ± standard deviation.

Incubation Time (h)	Inhibition (%)	
	Biomass	Cell Viability
24	99.4 ± 0.2	99.99 ± 0.01
48	91.0 ± 2.5	98.79 ± 0.17
72	60.8 ± 4.2	98.84 ± 0.23

## Data Availability

The datasets used and analyzed during the current study are available from the corresponding author on reasonable request.

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
