# Peer review of "Rhamnolipid 89 Biosurfactant Is Effective against Streptococcus oralis Biofilm and Preserves Osteoblast Behavior: Perspectives in Dental Implantology"

_ijms, 2023, doi:10.3390/ijms241814014_

Round 1
Reviewer 1 Report
The study concerning "Rhamnolipid 89 biosurfactant is effective against Streptococcus oralis biofilm and preserves osteoblast behavior: perspectives in dental implantology" is a comprehensive and interesting one.
My comments and suggestions are as follows:
Line 27. Please give full term for ATP.
Please use either Streptococus or S. throughout the text and abstract.
Line 95. Please use BS instead of the full term.
Materials and Methods should be section 2, not section 4. Please rearrange the sections. Results should be section 3, followed by discussion, section 4, and conclusions, section 5.
The materials should be described first, followed by the methods, please rearrange their sequence.
Line 147, 153, 201, 202, 220, 221, 224 etc. Please use figure instead of fig.
The discussion part is well written, and the conclusion is supported by the results.
Author Response
Ref.: Ms. No. ijms-2596583
Response to Reviewer # 1
We like to thank the reviewer for the evaluation of our manuscript and for the comments, which helped improving our manuscript.
Below we respond point by point to the various comments and indicate the changes we made in the revised manuscript. Revised text is highlighted in yellow in the new manuscript version.
Q: The study concerning "Rhamnolipid 89 biosurfactant is effective against Streptococcus oralis biofilm and preserves osteoblast behavior: perspectives in dental implantology" is a comprehensive and interesting one.
A: We thank the reviewer for the positive feedback.
Q: Line 27. Please give full term for ATP.
A: Acknowledged and changed. The abbreviation has been replaced with the full name. The same was applied to the ALP acronym defined at line 216.
Q: Please use either Streptococcus or S. throughout the text and abstract.
A: According to the journal guidelines, microorganisms’ name was indicated with full genus and species the first time they appear in the manuscript. Then, genus name was abbreviated using initial capital letter only.
Q: Line 95. Please use BS instead of the full term.
A: We thank the reviewer for this stylistic note. The term “Biosurfactant/s” has been abbreviated with “BS/s” throughout the whole manuscript, after being defined at line 58.
Q: Materials and Methods should be section 2, not section 4. Please rearrange the sections. Results should be section 3, followed by discussion, section 4, and conclusions, section 5.
A: The manuscript was arranged according to “Instruction for Authors” of IJMS journal, available at https://www.mdpi.com/journal/ijms/instructions, requiring sections in the following order: Introduction, Results, Discussion, Materials and Methods, Conclusions.
Q: The materials should be described first, followed by the methods, please rearrange their sequence.
A: Acknowledged. Subsections 4.1 and 4.2 were inverted to report first materials and then methods.
Q: Line 147, 153, 201, 202, 220, 221, 224 etc. Please use figure instead of fig.
A: Acknowledged and changed throughout the whole manuscript.
Reviewer 2 Report
Abstract. This section is adequate and show the summary of the paper.
Introduction.
This investigation is a paper that presents information for researchers in the field of biofilms in implant dentistry that can be responsible for severe associated diseases, such as mucositis and peri-implantitis, that affect the tissue around the implant. Microbial biosurfactants (BSs) are a group of natural molecules with strong antimicrobial, antiadhesive, and antibiofilm activity against a wide range of bacterias. When applied as coating agents making them less attractive to microbial colonization and adhesion. In several studies, BSs demonstrated the ability to prevent the formation of oral pathogens biofilms on commercial titanium surfaces against microbial colonization. Moreover, BSs showed the biocompatibility showed in titanium discs was non cytotoxic for both fibroblasts and osteoblasts.
The aim of the study is analize the safety and efficacy of R89BS biosurfactant as antibiofilm agent in dental implantology.
Materials and methods.
In experimental research, the section material and methods is very important for a scientific reproducibility.
This section explained the biosurfactant production according a rigorous methodology with related references.
In the subsection 4.2. Streptococcus oralis Growth Conditions, the authors must report some references for this experimental protocol.
In the subsection 4.3. Antibacterial Activity of R89BS on Planktonic Cells, the authors must report more references for this experimental protocol.
In the subsection 4.6. Medical-Grade Titanium Discs Preparation, the authors reported the preparation of discs with machined surface implant. This procedure is an important limitation of the study, because today, almost surface implants used are treated (i.e. sandblasting, etching) and consecutively, rough.
In the subsection 4.11. hOB adhesion and proliferation tests, the authors must report some references for this experimental protocol.
Results
This section reported the antibacterial activity of R89BS against S. oralis planktonic cells, also the rhamnolipid altered both cell surface hydrophobicity and membrane permeability, compared to S. oralis untreated cells. SEM imaging of TDs showed a markedly lower amount of bacterial cells adhering on R89BS-coated TDs with respect to uncoated TDs at all the investigated time points.
Moreover, the adhesion and proliferation behavior of hOBs on the surface of uncoated and R89BS-coated showed no significant differences.
Discussion.
In this study, the antimicrobial and antibiofilm activity of R89BS was evaluated against a different bacterial species, S. oralis, but the composition of human oral biofilm is more complex and included many bacterial species. Hence, this approach is a important limitation of the study.
The authors reported that these changes in attachment and growth of bacterial biofilm did not reflect into a detrimental interaction with eukaryotic cells. This finding is very important for the biologic safety of R89BS.
Some paragraphes included too information without any reference (i.e. line 344-355)
Conclusively, the study is not ready for publication.
Author Response
Ref.: Ms. No. ijms-2596583
Response to Reviewer # 2
We like to thank the reviewer for the evaluation of our manuscript and for the useful comments, which helped improving our research activity in this field.
Below we respond point by point to the various comments and indicate the changes we have made in the revised manuscript. Revised text is highlighted in yellow in the new manuscript version.
Q: Abstract. This section is adequate and show the summary of the paper.
A: We thank the reviewer for the positive feedback.
Q: Introduction. This investigation is a paper that presents information for researchers in the field of biofilms in implant dentistry that can be responsible for severe associated diseases, such as mucositis and peri-implantitis, that affect the tissue around the implant. Microbial biosurfactants (BSs) are a group of natural molecules with strong antimicrobial, antiadhesive, and antibiofilm activity against a wide range of bacterias. When applied as coating agents making them less attractive to microbial colonization and adhesion. In several studies, BSs demonstrated the ability to prevent the formation of oral pathogens biofilms on commercial titanium surfaces against microbial colonization. Moreover, BSs showed the biocompatibility showed in titanium discs was noncytotoxic for both fibroblasts and osteoblasts. The aim of the study is analize the safety and efficacy of R89BS biosurfactant as antibiofilm agent in dental implantology.
A: We thank the reviewer for summarizing the correct perspective of the study and its intended objectives.
Q: Materials and methods. In experimental research, the section material and methods is very important for a scientific reproducibility. This section explained the biosurfactant production according a rigorous methodology with related references. In the subsection 4.2. “Streptococcus oralis Growth Conditions”, the authors must report some references for this experimental protocol. In the subsection 4.3. Antibacterial Activity of R89BS on Planktonic Cells, the authors must report more references for this experimental protocol.
A: Several references were added to methodological subsections, including subsection 4.2 (now 4.1 in the revised manuscript) and subsection 4.3.
Q: In the subsection 4.6. Medical-Grade Titanium Discs Preparation, the authors reported the preparation of discs with machined surface implant. This procedure is an important limitation of the study, because today, almost surface implants used are treated (i.e. sandblasting, etching) and consecutively, rough.
A: We agree with the reviewer about the importance of testing antibiofilm efficacy on a range of different surface morphologies, including rough finishing such as those obtained by sand blasting, acid etching, laser ablation, and their combinations, as frequently applied in commercially available implants. Previous studies from Tambone et al. (doi:10.1186/s12903-021-01412-7) addressed this point using R89BS. Results showed that similar biofilm inhibition percentages were obtained, irrespectively on surface morphology, on three different commercial surface finishing, including both rough and smooth morphologies. However, the relevance of smooth titanium finishing should not be underestimated. Indeed, smooths finishing are widely used in coronal implant portions and transmucosal components, playing a key-role in favouring soft tissue integration, generation of mucosal seal and limiting microbial penetration towards apical areas of the implant.
Following the reviewer’s comment, the need to address efficiency of R89BS on different morphologies was added to the “Study Limitations” section.
Q: In the subsection 4.11. hOB adhesion and proliferation tests, the authors must report some references for this experimental protocol.
A: According to reviewer’s suggestion, additional references were added in subsections 4.9, 4.10 and 4.11. Bibliographic list has been updated accordingly.
Q: Results. This section reported the antibacterial activity of R89BS against S. oralis planktonic cells, also the rhamnolipid altered both cell surface hydrophobicity and membrane permeability, compared to S. oralis untreated cells. SEM imaging of TDs showed a markedly lower amount of bacterial cells adhering on R89BS-coated TDs with respect to uncoated TDs at all the investigated time points. Moreover, the adhesion and proliferation behaviour of hOBs on the surface of uncoated and R89BS-coated showed no significant differences.
A: We thank the reviewer for grabbing the key findings of our study.
Q: Discussion. In this study, the antimicrobial and antibiofilm activity of R89BS was evaluated against a different bacterial species, S. oralis, but the composition of human oral biofilm is more complex and included many bacterial species. Hence, this approach is an important limitation of the study.
A: We fully agree with the reviewer comment and this aspect was already addressed in the “Study Limitations” subsection 3.1. To make it clearer, we further elaborate and add some relevant references on this topic in the revised manuscript.
Q: The authors reported that these changes in attachment and growth of bacterial biofilm did not reflect into a detrimental interaction with eukaryotic cells. This finding is very important for the biologic safety of R89BS. Some paragraphs included too information without any reference (i.e. line 344-355). Conclusively, the study is not ready for publication.
A: We thank the reviewer for underlining the importance of the study findings. We revised the discussion section, and in particular the paragraph about biological safety results by adding punctual references to support our results and compare them with previous literature findings. Bibliographic list has been updated accordingly.
Round 2
Reviewer 2 Report
The review is correct